# Multimaterial 3D Printing for Arbitrary Distribution with Nanoscale Resolution

**DOI:** 10.3390/nano9081108

**Published:** 2019-08-02

**Authors:** Fengqiang Zhang, Changhai Li, Zhenlong Wang, Jia Zhang, Yukui Wang

**Affiliations:** 1School of Mechatronics Engineering, Harbin Institute of Technology, Harbin 150001, China; 2Key Laboratory of Micro-systems and Micro-structures Manufacturing, Ministry of Education, Harbin Institute of Technology, Harbin 150080, China

**Keywords:** 3D printing, nano-resolution, arbitrary distribution, multimaterials, deposition surface, rapidity, large scale

## Abstract

At the core of additive manufacturing (3D printing) is the ability to rapidly print with multiple materials for arbitrary distribution with high resolution, which can remove challenges and limits of traditional assembly and enable us to make increasingly complex objects, especially exciting meta-materials. Here we demonstrate a simple and effective strategy to achieve nano-resolution printing of multiple materials for arbitrary distribution via layer-by-layer deposition on a special deposition surface. The established physical model reveals that complex distribution on a section can be achieved by vertical deformation of simple lamination of multiple materials. The deformation is controlled by a special surface of the mold and a contour-by-contour (instead of point-by-point) printing mode is revealed in the actual process. A large-scale concentric ring array with a minimum feature size below 50 nm is printed within less than two hours, verifying the capacity of high-throughput, high-resolution and rapidity of printing. The proposed printing method opens the way towards the programming of internal compositions of object (such as functional microdevices with multiple materials).

## 1. Introduction

Three-dimensional printing is well-known as one of the disruptive technologies affecting the future, which may bring about the third industrial revolution. Due to its capacity of rapid prototyping for arbitrary shape, it is widely applied in aerospace [1], medical health [2,3], structural electronics [4], and mold [5]. The traditional 3D printing technologies including stereolithography [6,7], selective laser sintering [8,9], fused deposition modeling [10,11] have consistently been improved, however they are still subject to the low-throughput and low-resolution fundamentally resulting from point-by-point printing and large printing unit size, in which the optimum printing resolution of approximately 20–50 μm is available [12]. Therefore, as an extension of 3D printing, continuous liquid interface production is developed with higher efficiency [13,14,15], as well as femtosecond laser direct writing based on two-photon polymerization with higher resolution of up to 100 nm [16,17,18,19,20,21]. Nevertheless, a trade-off of efficiency and resolution is still difficult to realize.

In addition, without an effective and precise controlling strategy of position of multiple materials, the state-of-the-art 3D printing technology still faces enormous challenges to achieve real printing of multiple materials for arbitrary distribution [22,23,24,25,26,27,28]. The study of 3D printing has been stagnating at the primary stage of controlling object shape for a long time [6] and a serious breakthrough is needed for the senior stage of programming internal compositions [29], despite some achievements such as the printing with hybrid multimaterials [22], printing via multiple nozzles with different materials [23,30], simple and imprecise controlling of compositions by the assisted physical morphology [24,25,26], magnetism [27], chemical reaction [28], etc. Three-dimensional printing of multiple materials with nano-resolution can create further applications in micro- and nano-scale multifuctional device [31,32] and high-level biological tissues [33]. Therefore, how to control multiple materials with high resolution and efficiency to construct objects has attracted more interest from the researchers. As mentioned by American scientist Hod Lipson, the core of real 3D printing was the capacity of achieving arbitrary distribution of multiple materials [29], which is what we have been doing.

In this study, we demonstrate the precise control of the position of multiple materials for arbitrary distribution by means of the morphological characteristics. The established physical model shows that a complex distribution on a section can be achieved by the vertical deformation of multimaterial lamination. The deformation is controlled by a special surface of the mold and a contour-by-contour printing mode is revealed in the actual process. A large-scale concentric ring array with a minimum feature size below 50 nm is printed within less than two hours, demonstrating the nano-resolution and rapidity of printing. Simultaneous improvement of resolution and efficiency considered as a pair of contradictions is realized. The proposed printing strategy makes a great extension of function and application of 3D printing technologies.

## 2. Experimental Details

### 2.1. Fabrication of Concentric Ring Array

A 2-inch patterned sapphire substrate (PSS, Zhejiang Bolant Semiconductor Technology Co., Ltd., Zhejiang, China.) with conical structure array (Appendix A) was used as the deposition substrate and cleaned by argon (Ar) plasma before deposition of materials. Then the alternative depositions of tungsten and aluminium (99.99% purity, Beijing Goodwill Metal, Co., Ltd., Beijing, China) on the PSS were implemented in the argon atmosphere of 0.45 Pa using a high vacuum multiple target magnetron sputtering coating machine (JCP-350M2, Beijing Technol Science Co., Ltd., Beijing, China). Deposition thickness of W and Al was set at 120 nm except the first layer with thickness of 240 nm, and the total number of deposition layers was twenty-three (See Appendix A for details). To achieve uniform thickness of deposition film, the substrate holder was simultaneously cooled with circular liquid nitrogen and rotated during deposition. Finally, the required lamination was obtained (Appendix A) and the concentric ring array on the section *z* = 1.83 μm of lamination was further revealed after removing the upside materials using ultra-precision lathing and polishing paste (Al_2_O_3_, 10–50 nm, Microspectrum Technology Co., Ltd., Shanghai, China.). The whole process took less than 2 h.

### 2.2. Characterization

The scanning electron microscope (SEM) images were collected from the field emission scanning electron microscope (Supra 55 Sapphire, Carl Zeiss, Oberkochen, Germany) at an acceleration voltage of 20 kV. The morphology of microstructures array was measured by the atomic force microscope (AFM, dimension fastscan, Bruker, Billerica, MA, USA). The transmission electron microscopy (TEM) specimen was prepared by using Helios NanoLab 600i FIB/SEM Dual-Beam system (FEI, Hillsboro, OR, USA). The TEM, high-resolution TEM (HRTEM), energy-dispersive X-ray (EDX) and high-angle annular dark field-scanning transmission electron microscopy (HAADF-STEM) images were taken from in-situ multifunctional transmission electron microscopy (Talos F200X, FEI, Hillsboro, OR, USA) at an acceleration voltage of 200 kV.

## 3. Results and Discussion

### 3.1. 3D Printing Strategy for an Arbitrary Distribution of Multiple Materials

Figure 1 illustrates the 3D printing strategy for an arbitrary distribution of multiple materials. The principle diagram of achieving arbitrary distribution is shown in Figure 1a–d. The lamination *S* is composed of materials *W*(*i*) (*i* = 1 … *n*) in a certain order, in which an arbitrary curved surface *z* = *f*(*x*, *y*) and datum *z* = *g* are given (Figure 1a). Deformation from the surface *z* = *f*(*x*, *y*) to the datum *z* = *g* is implemented (Figure 1b). For instance, these points I–VII on the surface *z* = *f*(*x*, *y*) move vertically *h*_I_–*h*_VII_ to the datum, respectively. Correspondingly, the transformation from the lamination *S* to lamination *V* is forced as well as the bottom plane *z* = *k* to the surface *z* = *F*(*x*, *y*) by the same rule. The distribution of multiple materials is obtained by intercepting the section *z* = *g* in the lamination *V* (Figure 1c). In addition, the curved surface *z* = *f*(*x*, *y*) is endowed with corresponding materials and projected vertically onto a plane, resulting in the transverse distribution (Figure 1d). The comparison reveals that the distribution on the section *z* = *g* is absolutely the same with the transverse distribution of the curved surface *z* = *f*(*x*, *y*) within the lamination *S*. Therefore, an arbitrary and complex distribution on a plane can be programmed and achieved via the designing of corresponding curved surface *z* = *f*(*x*, *y*) and longitudinal deforming of lamination *S* of multiple materials. The core of the strategy lies in the realizing of transformation from simply longitudinal laminating of multiple materials to complex transverse distribution via longitudinal deformation.

Based on the principle, the schematic diagram of 3D printing process for arbitrary distribution of multiple materials is revealed, in which the longitudinal deformation is controlled by the morphology (Figure 1e–l). According to the bottom surface *z* = *F*(*x*, *y*) of lamination *V* (Figure 1b), the deposition surface *z* = *F*(*x*, *y*) of the mold is prepared and source materials is given for 3D printing (Figure 1e). Then the deposition on the deposition surface is conducted in order of *W*(1)…*W*(*i*)…*W*(*n*) through atomic deposition technology such as magnetron sputtering (Figure 1f–k, top), resulting in controllable printing of corresponding material on the section *z* = *g* (Figure 1f–k, bottom) and the lamination *V* can be achieved (Figure 1k). Finally, the designed distribution of multiple materials is obtained by removing these materials upon the section *z* = *g* (Figure 1l). Notably, deposition on the deposition surface is obviously the process of printing multiple materials for the distribution on the section *z* = *g*, and the deposition rate determines the printing speed. Contour-by-contour printing of materials is conducted according to these section contours of deposition surface from top to bottom (Figure 1f–k, bottom), in which the dependence of printing direction is on the height of morphology. The contour-by-contour (rather than point-by-point) printing brings about the rapidity of 3D printing of multiple materials.

Figure 2 illustrates the analysis of the printing resolution. The lamination *V* is achieved via multimaterial deposition on a conical structure with triangular contour, in which the slope of the mold and the thickness of deposition layer *W*(*i*) are α and *h*, respectively (Figure 2a). Then the width *w* of *W*(*i*) on the section *z* = *g* in the lamination *V* is investigated. There is an equation *w* = *h*/tan(α) resulting in the width *w* of *W*(*i*) decreasing with the decrease of the thickness *h* of *W*(*i*) or the increase of the slope α of the mold. The width *w*_1_ and *w*_2_ of *W*(*i*) on the section *z* = *g* are obtained by decreasing the thickness (*h*_1_ < *h*) of deposition layer *W*(*i*) and increasing the slope (α_2_ > α) of the mold, resulting in the inequalities *w*_1_ < *w* and *w*_2_ < *w*, respectively (Figure 2b,c). From Figure 2, the width *w* of *W*(*i*) on the section *z* = *g* can theoretically reach nanoscale and even atomic level, when the thickness *h* of *W*(*i*) is small enough or the slope α of the mold is lager enough. Notably, the thickness of the monolayer atoms can be easily fabricated by the atomic deposition. Therefore, nanoscale resolution printing is feasible and suitable for the substrate with a complex structure. At this time, a good trade-off between the resolution and throughput is realized. 

In the process of 3D printing, the deposition surface acts as a series of masks, and contour-by-contour printing of same-material on the section *z* = *g* is conducted to further achieve self-assembly of multiple materials. The distribution on the section *z* = *g* is actually a mapping of the deposition surface and mapped by laminating multiple materials on the deposition surface. Therefore, this is a new method of 3D printing for rapidly achieving an arbitrary nano-resolution distribution of multiple materials on a large scale, which is called Morphology-Guided Printing (MGP).

### 3.2. Concentric Ring Array Fabricated by MGP Method

Figure 3a–f schematically illustrates the MGP process of multiple materials on the printing section *z*=*g* based on the PSS with conical structure array (Appendix A). Half of a single conical structure is used for observing and has a like-parabola contour. The pattern on the section *z* = *g* is printed circle by circle from center to edge via the deposition of multiple materials. Using the MGP method, a concentric ring array is fabricated via alternative deposition of tungsten and aluminium within less than two hours for investigation (Figure 3g–j and Appendix A). Figure 3g,h presents top and side view SEM images of pre-processing multimaterial lamination with a thickness of 4.32 μm, respectively. Honeycomb contour and good uniformity of microstructures (Appendix A) of lamination are observed, and uniformity of thickness of deposition layer and alternative laminating of tungsten (white) and aluminium (black) are demonstrated by the internal structure of lamination (red dotted box in Figure 3h). This demonstrates good capacity of deposition of materials and provides strong support for high-solution printing of multiple materials on a certain section. However, some bubbles on the lamination are observed, which may be the result of atomic aggregation in the process of deposition and affect printing accuracy of materials. Figure 3i,j reveals top and side SEM images of post-processing lamination of materials, respectively. The section *z* = 1.83 μm is obtained via removing upper materials, and the uniform concentric ring array is further developed. However, the insert of Figure 3i shows tortuous boundary of rings, resulting from these bubbles. For the section *z* = 1.83 μm of single microstructure, sequential same-material rings are printed circle by circle from center to edge and at the center of the section is just aluminum (red dotted circle shown in Figure 3j). Generally, a large-scale distribution of multiple materials can be rapidly printed on the specified section.

To further investigate the printing capacity of MGP method for arbitrary distribution of multiple materials, the section *z* = 1.83 μm with a thickness of less than 100 nm is intercepted by focused ion beam (FIB) and characterized by TEM (Figure 4a–f). Figure 4b shows HAADF-STEM image of single concentric rings unit (red dotted box of Figure 4a), which demonstrates W (white)-and-Al (black) alternative concentric rings. Figure 4c–e exhibits the EDX elemental mapping images of Al, W and their overlap, respectively, which further verify W-and-Al alternative concentric distribution. Figure 4f reveals Al/W component distribution curves along the purple line shown (shown in Figure 4b), in which the position of zero coordinate presents the center of the concentric ring. The transforming of like-rectangle to triangle wave is accompanied by the decreasing of the period in the radial direction. This indicates the widths of the concentric rings have a decreasing trend with increasing in radius, which is caused by the increasing slope of the conical structure (Appendix A). And the W content of less than 100% indicates the doping in W region. Notably, the doping may be caused by atomic diffusion or environmental pollution in the process of deposition. For quantifying the distribution size of materials, the TEM image of specimen is processed using MATLAB software to get ten concentric nanorings with clear and tortuous boundaries (Figure 4g). Then the mean radii and width of distribution of materials are calculated (Table 1), further verifying the decreasing trend of the ring widths. The emphasis is on that the smallest feature size of 41.8nm demonstrates the nanoscale resolution of MGP method.

Then the crystal analysis on the section *z* = 1.83 μm is conducted (Figure 5). Figure 5a shows the TEM image of specimen, where the white and black regions represent the materials of Al and W, respectively. Then the corresponding HRTEM images of these regions marked with red boxes shown in Figure 5a and their selected area electron diffraction (SAED) patterns are exhibited in Figure 5b–i top and bottom, respectively. The evolution of crystal state is shown in Figure 5b–e, in which Figure 5b displays the good single crystal nature of Al in the pure Al region, the crystals of W begin to appear with the doping of W in the transition region (Figure 5c,d) and Figure 5e exhibits the amorphous nature of W in pure W region. Additionally, the red curves (shown in Figure 5c,d) are the boundaries between pure Al, transition and pure W regions. It is worth noting that the single crystals nature of Al is not destroyed by a small amount of doped W in transition region (Figure 5c,d). However, the poly-crystallization occurs in other regions such as transition (Figure 5f,h), pure Al (Figure 5g), and W (Figure 5i) regions, induced by the doping of other elements. This further results in refinement of crystals, low crystallinity and diversification of crystal orientation. Generally, a small amount of doping in the process of printing is further revealed by the crystal analysis, and the same-material printing region of small enough size is vulnerable to contamination. Meanwhile, the doping can be eliminated by some measures, such as using the targets of higher purity, prevention of mutual contamination of targets, improved cooling system for restraining atomic diffusion, and maintaining the adequate cleanliness of the sputtering chamber by cleaning up impurities and an improved vacuum system.

In addition, using the same substrate (such as PSS), other distributions are available by programming the material type, sequence and thickness of each deposition layer. For instance, the radial gradient circle can be printed by depositing gradient materials on the PSS (Figure 3a–f). Furthermore, when the substrate with a different morphology is used as the substrate, the corresponding distribution can be achieved via the design and depositing of multiple materials. Appendix A exhibits a printed distribution on the specified section via depositing gradient materials on the morphology of monolayer cells. By comparison, a high consistency between SEM image of the distribution and optical image of the morphology is observed. Therefore, the feasibility and high resolution of MGP method is further validated as well as the diversity of distribution of multiple materials.

## 4. Summary

In summary, a new method of nano-resolution 3D printing for rapidly achieving arbitrary distribution of multiple materials on a large scale is proposed. The designed distribution of multiple materials on the specified section can be obtained by the corresponding deformation of lamination of multiple materials. In fact, the deformation of lamination can be realized via the corresponding morphology. Based on atomic deposition on PSS, a concentric ring array with nanoscale resolution on a large scale is printed verifying the rapidity and nanoscale resolution of MGP method. The improvement of atomic deposition including collimation and secondary deposition, and the decreasing of thickness of each layer, will contribute to higher and even atomic resolution of MGP. In addition, almost all materials can be used and are mutually compatible for MGP due to the abundant deposition methods including physical vapor deposition (PVD) and chemical vapor deposition (CVD). Versatile MGP method is also suitable for particle deposition to print macro-scale architecture. More importantly, these sections with the designed distribution can be isolated by micro–nano processing technology such as FIB and further laminated to form an arbitrary distribution of multiple materials in three-dimensional space. Therefore, MGP method makes a great extension of function and application of additive manufacturing and opens the way towards the programming internal components of objects such as transistors.

## Figures and Tables

**Figure 1 nanomaterials-09-01108-f001:**
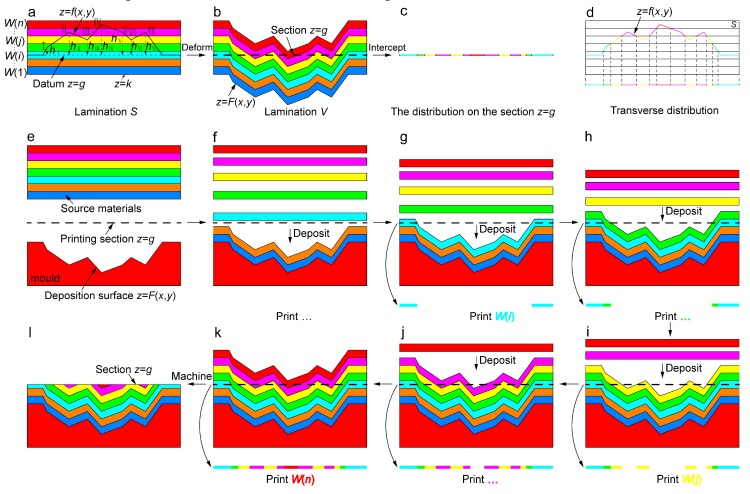
Three-dimensional printing strategy for arbitrary distribution of multiple materials. (**a**–**d**) Principle diagram of achieving arbitrary distribution of multiple materials. Arbitrary curved surface *z* = *f*(*x*, *y*) and datum *z* = *g* are given within lamination *S* of multiple materials *W*(*i*) (**a**). The deformation (**b**) from the surface *z* = *f*(*x*, *y*) to the datum *z* = *g* is implemented. For instance, the points I–VII of surface *z* = *f*(*x*, *y*) move vertically *h*_I_–*h*_VII_ to the datum, respectively. Correspondingly, the transformation from lamination *S* to lamination *V* is forced by the same rule. Then the section *z* = *g* (**c**) is intercepted from lamination *V*, on which the multimaterial distribution can be obtained. That the distribution on the section *z* = *g* is exactly the same with transverse distribution of surface *z* = *f*(*x*, *y*) (**d**) is observed. (**e**–**l**) Schematic diagram of 3D printing process for arbitrary distribution of multiple materials. The deposition surface of the mold is prepared and source materials *W*(*i*) (*i* = 1 … *n*) are designed for printing (**e**). Materials *W*(1)…*W*(*i*)…*W*(*n*) are successively deposited on the deposition surface of the mold to further print corresponding material on the section *z* = *g* (**f**–**k**). The section *z* = *g* can be obtained by machining deposition body (**l**).

**Figure 2 nanomaterials-09-01108-f002:**
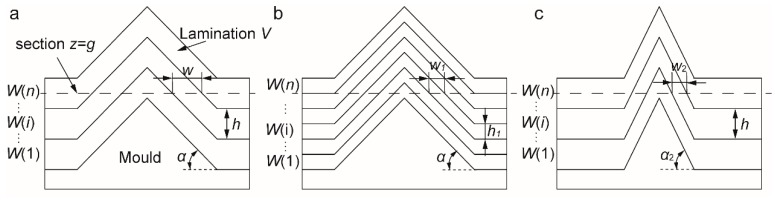
Analysis of the printing resolution. (**a**) The width *w* of *W*(*i*) on the section *z* = *g* in the lamination *V* under the conditions of the slope α of the mold and the thickness *h* of deposition layer *W*(*i*). There is an equation *w* = *h*/tan(α) resulting in the width *w* of *W*(*i*) decreasing with the decrease of the thickness of *W*(*i*) or the increase of the slope of the mold. (**b**) The width *w*_1_ of *W*(*i*) on the section *z* = *g* by decreasing the thickness (*h*_1_ < *h*) of deposition layer *W*(*i*). There is an inequality *w*_1_ < *w*. (**c**) The width *w*_2_ of *W*(*i*) on the section *z* = *g* by increasing the slope (α_2_ > α) of the mold. There is an inequality *w*_2_ < *w*.

**Figure 3 nanomaterials-09-01108-f003:**
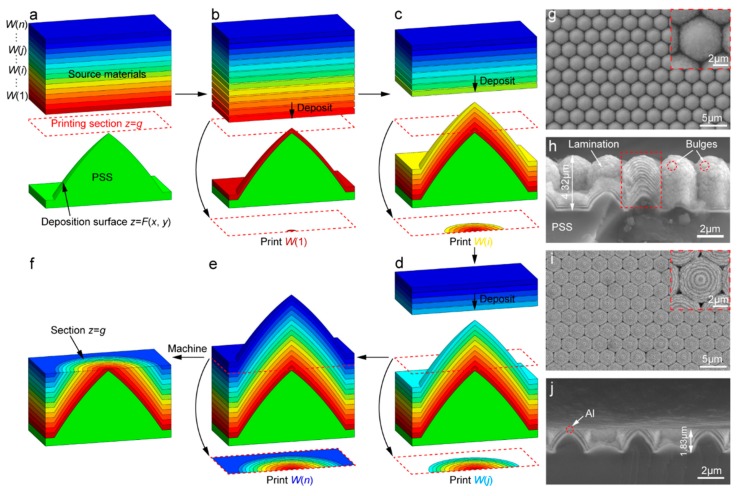
Morphology-Guided Printing (MGP) of multiple materials on printing section *z* = *g* based on patterned sapphire substrate (PSS). (**a**–**f**) Schematic diagram of MGP process. PSS and source materials *W*(*i*) (*i* = 1…*n*) for printing are given (**a**). Materials *W*(1)…*W*(*i*)…*W*(*n*) are successively printed circle by circle on the section *z* = *g* (**b**–**e**). The section *z* = *g* is obtained by machining (**f**). (**g**–**j**) MGP of concentric circle array. Top (**g**) and side (**h**) view SEM images of pre-processing multimaterial lamination with a thickness of 4.32 μm are revealed on the PSS. Lamination is formed via alternative depositing of tungsten and aluminum. Honeycomb contour of lamination and good uniformity of microstructures of lamination is demonstrated in (**g**) and a single microstructure is revealed in the insert. W (white)-and-Al (black) alternating lamination (red dotted box) and bulges on the surface of lamination (red dotted circle) are observed in (**h**). Top (**i**) and side (**j**) view SEM images of post-processing lamination are shown. The section *z* = 1.83 μm is obtained via removing materials. Uniform concentric circle array is shown, and a single concentric circle is revealed in the insert. At the center of the section is just aluminum.

**Figure 4 nanomaterials-09-01108-f004:**
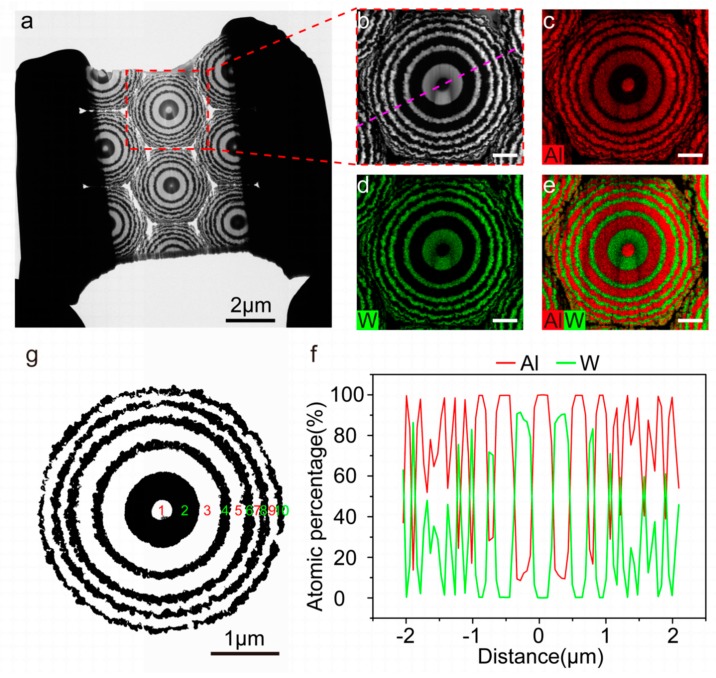
Elemental analysis of single concentric ring unit. (**a**) TEM image of specimen. The section *z* = 1.83 μm with W-and-Al alternating rings printed is intercepted by FIB and the thickness of specimen is less than 100 nm. (**b**) HAADF-STEM image of single concentric ring unit shown in red dotted box of (**a**). EDX elemental mapping images of Al (**c**), W (**d**), and their overlap (**e**) are shown. (**f**) Material distribution curves along the purple dotted line shown in (**b**). Scale bar: 500 nm in (**b**–**e**). (**g**) Post-processing material distribution including ten concentric rings using MATLAB software. Material distribution is extracted from TEM image of specimen (**a**) using image processing technology and clearer boundaries are shown. Al (white)-and-W (black) alternative concentric rings are labeled 1 to 10 in turn.

**Figure 5 nanomaterials-09-01108-f005:**
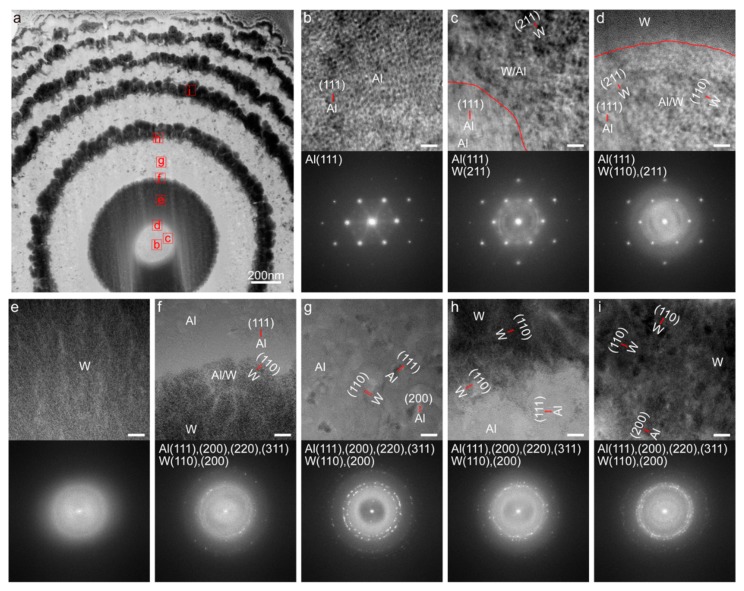
Crystal analysis of specimen. (**a**) TEM image of specimen. (**b**–**i**) HRTEM images (top) and corresponding SAED patterns (bottom) of these regions marked with red boxes in (**a**). Scale bar: 10 nm in (**b**–**i**).

**Table 1 nanomaterials-09-01108-t001:** Radius and width of distribution of multiple materials in Figure 4g.

Concentric rings ^a^	1	2	3	4	5	6	7	8	9	10
Circle radius ^b^ (nm)	65.7 ± 9.5	228.3 ± 8.6	372.3 ± 12.4	424.9 ± 11.4	524.4 ± 16.5	578.2 ± 14.7	623.9 ± 16.0	668.4 ± 12.7	710.6 ± 14.4	752.4 ± 13.7
Ring width ^c^ (nm)	65.7	162.6	144	52.6	99.5	53.8	45.7	44.5	42.2	41.8

^a^ The concentric rings marked 1 to 10 are shown in Figure 4g. ^b^ Presenting the mean radius and error of outer circle of each ring calculated by MATLAB software. ^c^ Determined by the difference between outer and inner radius of each ring.

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
