# Peer review of "Multimaterial 3D Printing for Arbitrary Distribution with Nanoscale Resolution"

_nanomaterials, 2019, doi:10.3390/nano9081108_

Round 1

Reviewer 1 Report

In my opinion only some minor linguistic mistakes should be corrected and the article could be accepted. Congratulations to Authors for very well prepared manuscript. 

Reviewer 2 Report

The author of this manuscript developed a new method of nano-resolution 3D printing for rapidly achieving arbitrary distribution of multiple materials on a large scale. However, it is not clear how to fabricate the nano-scale resolution printing. The author should modify the manuscript for understanding their strategy easily. And, the results were not clearly presented. Therefore, I recommend to reject this manuscript in Nanomaterials.

1.    For using this method, it looks like that pattern sapphire substrate (PSS) with conical structure array should be required. How to prepare the pattern sapphire substrate (PSS) with conical structure array? If we want to use this method for fabricating large-scale area, how to prepare the PSS substrates in large-scale?

2.    In this method, it looks like that we can only fabricate the concentric ring array. Can we fabricate the various shape patterns in large-scale?

3.    For 3D printing, vertically accumulation is also important. How to fabricate 3D structures with nano-scale resolution?

Reviewer 3 Report

Authors show an alternative route to grow complex structures, using traditional microelectronic techniques, such as sputtering, lathing and polishing. They launch this procedure as a new 3d printing method, which exhibits a nanoscale resolution.

I regret to say that such a process cannot be described as 3d printing method, since the final result is not a 3 dimensional printed structure.  The traditional additive manufacturing methods are used to obtain 3 dimensional structures. In contrast, the final concentric circle array, obtained using the proposed procedure, is a pure 2D structure, with well-defined characteristics.

Although, the proposed method is indeed interesting, especially due to the fact that multiple materials can be used, towards the development multimaterial, complex structures.  Furthermore, by using other substrates, rather than cones, structures with other shapes can be achieved. The authors could discuss on that issue, or, even better, they can show experimental evidence, if they have.

In conclusion the paper is interesting, the results are remarkable, however the procedure cannot be considered as a new 3D printed method. For this reason the paper cannot be published.

Reviewer 4 Report

-"Generally, a small amount ... designed distribution.": The statement is considered rather generic, authors should provide evident information and clearly describe how this issue was tackled.

-Errors should be included in values of Table 1

-focused iron beam (FIB): should be revised

-"Specially, Deposition on the deposition surface": should be revised

Round 2

Reviewer 2 Report

The author was performed the appropriate experiments. The results were clearly presented and the manuscript was well organized and written. Therefore, I recommend to accept this manuscript in the nanomaterials.

Reviewer 3 Report

Authors did respond appropriately to my comments/suggestions, as well as they correspondingly revised the manuscript. Therefore, the manuscript can be considered as publishable to the Nanomaterials journal.